# Tropical cyclone rainbands can trigger meteotsunamis

Luming Shi [1]*, Maitane Olabarrieta [1]*, David S. Nolan[2]* & John C. Warner[3]*

Tropical cyclones are one of the most destructive natural hazards and much of the damage and casualties they cause are flood-related. Accurate characterization and prediction of total water levels during extreme storms is necessary to minimize coastal impacts. While meteotsunamis are known to influence water levels and to produce severe consequences, their impacts during tropical cyclones are underappreciated. This study demonstrates that meteotsunami waves commonly occur during tropical cyclones, and that they can contribute significantly to total water levels. We use an idealized coupled ocean–atmosphere–wave numerical model to analyze tropical cyclone-induced meteotsunami generation and propagation mechanisms. We show that the most extreme meteotsunami events are triggered by inherent features of the structure of tropical cyclones: inner and outer spiral rainbands. While outer distant spiral rainbands produce single-peak meteotsunami waves, inner spiral rainbands trigger longer lasting wave trains on the front side of the tropical cyclones.

[1] Civil and Coastal Engineering Department, ESSIE, University of Florida, Gainesville, FL 32611, USA. [2] Rosenstiel School of Marine and Atmospheric Science, University of Miami, Miami, FL 33149, USA. [3] US Geological Survey, Woods Hole Coastal and Marine Science Center, Woods Hole, MA 02543, USA. *email: shiluming@ufl.edu; maitane.olabarrieta@essie.ufl.edu; dnolan@rsmas.miami.edu; jcwarner@usgs.gov

Coastal flooding and erosion from extreme storms represent some of the main threats faced by coastal communities. Several recent hurricanes striking the United States have produced devastating consequences. Collectively, Hurricanes Michael (2018), Florence (2018), Maria (2017), Irma (2017), and Harvey (2017) resulted in 3269 direct and indirect deaths and estimated cost of $325 billion dollars[1]. As human populations along the coast continue to increase and sea levels rise, tropical cyclones (TCs) will likely result in enhanced coastal impacts in the future[2–4]. Therefore, having accurate and reliable storm-impact prediction models will be even more vital in the coming decades.

Despite ongoing scientific and computational advances, accurate prediction of coastal water levels during TCs is still a major challenge. This is mainly the result of the high uncertainty of the forecast atmospheric fields used to force hydrodynamic models, the scarcity of measurements during extreme storms, and the nonlinearity of the interactions between the different processes affecting total water levels. In the nearshore region, total water levels are known to be the result of astronomical tides, storm surges, sea-swell wave setup, water level changes due to infragravity waves, gravity wave runup, swash motions, and seasonal and climatic variations. The importance of rainfall has also become evident in recent hurricanes (e.g., Harvey, 2017; Florence, 2018). Regardless of the observational evidence that TCs and remnants of TCs have the potential to generate meteotsunamis[5–7] that can add to total water levels, their frequency and triggering and propagation mechanisms have not been analyzed yet. Moreover, these types of water level oscillations are usually not considered when forecasting and assessing TC impacts in coastal regions.

Meteotsunamis are tsunami-like sea level oscillations with periods from minutes up to several hours[8]. These multi-resonant ocean waves are initiated by moving atmospheric disturbances (sudden atmospheric pressure and/or wind changes) and are usually associated with frontal passages, squalls, thunderstorms, and atmospheric gravity waves. While meteotsunamis have been reported worldwide[8–10], and their impacts can be severe locally[9,10], there is still a vast gap in our knowledge about how frequently these types of water level oscillations occur and about their generation and propagation mechanisms, especially during TCs. Due to the coarse spatial (≥3 km) and temporal resolutions (≥1–3 h) of the atmospheric fields usually available to force storm surge models, current forecasting systems used to simulate water levels during TCs are ill-equipped to capture and model meteotsunamis.

Within the structure of TCs, tropical cyclone rainbands (TCRs) and atmospheric gravity waves are the features with the greatest potential to trigger meteotsunamis. TCRs represent the regions of heaviest precipitation outside the eyewall of the TC and, depending on the degree of influence by the inner-core vortex dynamics, are classified into inner (which include principal and secondary rainbands) and distant rainbands (often referred to as outer rainbands). The recent study by Yu et al.[11] showed that there is a high probability (>50% of the analyzed cases) of outer TCRs developing squall-line-like characteristics with the potential for severe weather conditions. These squall-line-like features are characterized by convective precipitation, ocean surface cold pool signatures, and wind convergence zones at low levels in the atmosphere. TCs are also known to produce atmospheric gravity waves in the stratosphere[12] and in the lower troposphere[13]. Although all these features of TCs could produce meteotsunamis, it is unknown which of them have the greatest potential to do so, and under what conditions this potential is maximized.

Here, we analyze water level, wind, atmospheric pressure, air temperature, and Next Generation Weather Radar (NEXRAD)

reflectivity observations between years 1998–2018 in the East Coast of the United States, the Northern Gulf of Mexico and Puerto Rico. The main goals are to explore how frequently meteotsunamis are triggered by TCs and to ascertain the main atmospheric structures responsible for their generation. We further explore the relationship between TC structure and meteotsunami generation and propagation by applying a coupled ocean–atmosphere–wave model to an idealized TC.

## Results

**Description of the considered TCs.** Based on the historical reports of the National Hurricane Center, from the 295 TCs of the 1998–2018 Atlantic hurricane seasons, we considered those that made landfall or propagated along the continental shelf of the Gulf of Mexico, Eastern United States, and Puerto Rico; these are 97 TCs in total. Forty-six of these TCs became hurricanes and 24 of them became major hurricanes during their lifetimes; 48 propagated through the Gulf of Mexico, 29 through the Atlantic coast/shelf and 20 through the Gulf of Mexico and the Atlantic coast/shelf.

For the duration of each TC, we analyzed all available water level, sea level atmospheric pressure, sustained wind speed, wind gust, and air temperature measurements from the National Oceanic and Atmospheric Administration (NOAA) tide gauges and meteorological stations (along the Gulf of Mexico, Eastern United States and Puerto Rico). We used the predicted tides from NOAA and a Lanczos filter (described in the Methods section) to isolate water level oscillations in the meteotsunami frequency band (hereafter referred to as water level anomaly) and the surge from the total water levels. We also applied this procedure to the atmospheric measurements. Four TCs (Leslie 2000, Charley 2004, Wilma 2005, and Gordon 2018) triggered fast surge variations concurrent with water level oscillations in the meteotsunami frequency band. With these fast changes in the surge levels, the filtering technique used in this study was unable to separate the surge from the anomalies in the meteotsunami frequency band. For this reason, these four specific events have been disregarded in the present analysis.

**Meteotsunami observations during TCs.** In 49 of the 93 analyzed TCs (52% of the cases), maximum meteotsunami elevations greater than 0.2 m were measured by at least one tide gauge. Nineteen events (20% of the cases) triggered maximum elevations higher than 0.3 m, and eight of them (8%) produced meteotsunami elevations above 0.4 m (Fig. 1a). The seven out of eight of these most extreme meteotsunami events occurred in the Gulf of Mexico (Fig. 1b) and were measured in the right-front and right-rear of the TCs.

Hurricane Harvey (2017) produced the largest observed meteotsunami: a single-peak meteotsunami wave with a maximum elevation of 0.78 m at Freshwater Canal Locks (Louisiana) (Fig. 1c). Hurricane Humberto (2007) produced persistent meteotsunami wave trains reaching elevations of 0.55 m at Freshwater Canal Locks, before and after the TC landfall. The first peak occurred at 1700 UTC September 12. A second major peak was measured 2 h later, at 1905 UTC. On September 13, two main peaks were observed at 1110 and 1420 UTC, respectively. Hurricane Gustav (2008) also triggered meteotsunamis at this particular tide gauge. The main peak with a maximum elevation of 0.49 m hit Freshwater Canal Locks a day ahead of the first TC landfall. Hours before and after the landfall of Gustav (2008), smaller trains of meteotsunami waves were detected at this tide gauge.

Hurricane Irma (2017) produced maximum meteotsunami elevations of 0.50 m at Trident Pier (Florida) 8 h before the

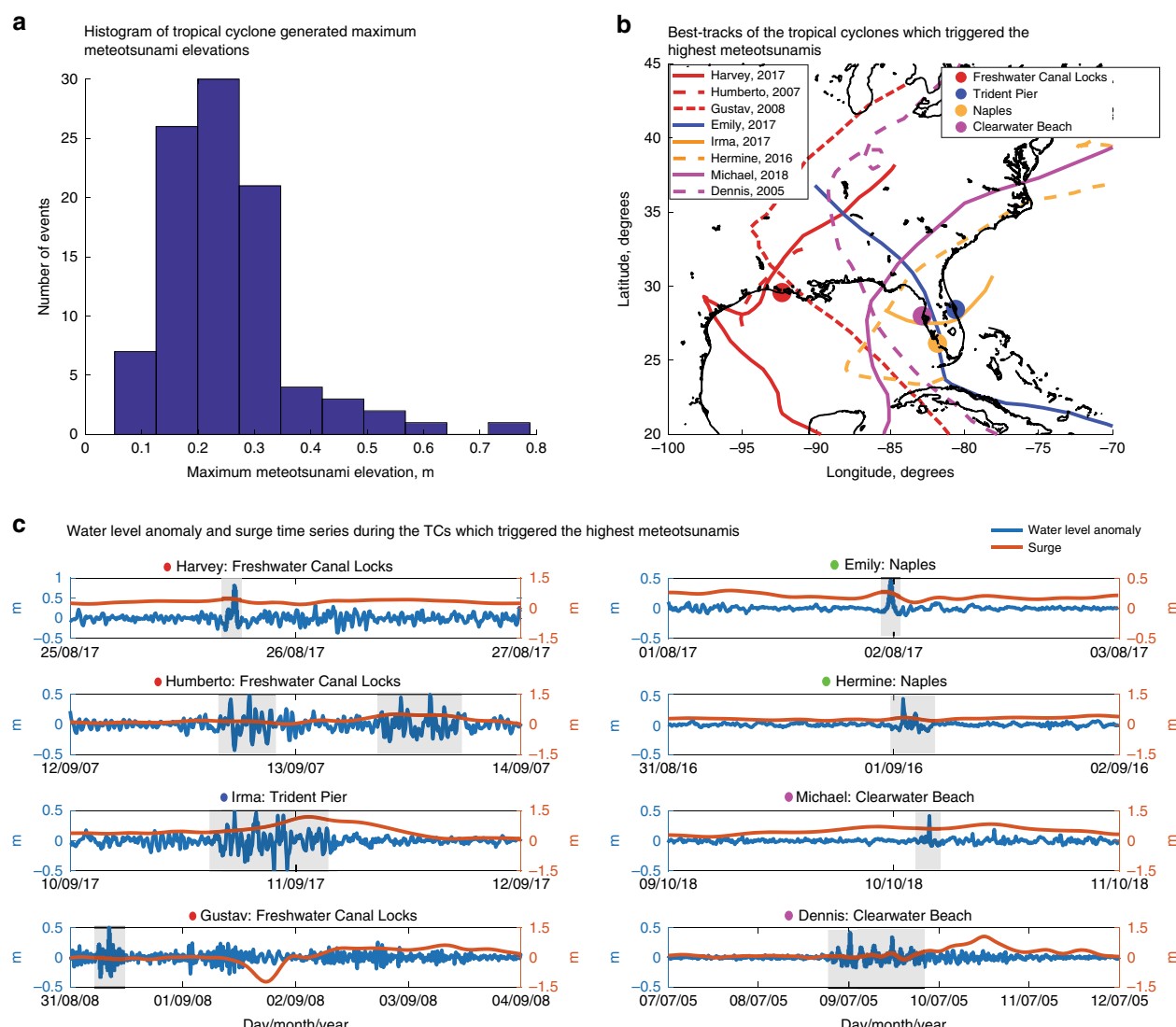

**Fig. 1 Tropical cyclone-induced meteotsunamis. a** Histogram of maximum meteotsunami elevations for the tropical cyclone (TC)-triggered meteotsunamis during the 1998–2018 Atlantic hurricane seasons. **b** Best-tracks of the TCs that created the eight maximum meteotsunami events. The red/gold/blue/magenta lines represent best-tracks of the TCs that created the maximum meteotsunami elevations in Freshwater Canal Locks/Naples/Trident Pier/Clearwater Beach respectively. The locations of Freshwater Canal Locks/Naples/Trident Pier/Clearwater Beach tide gauges are indicated with red/gold/blue/magenta dots, respectively. **c** Storm surge and water level anomaly (period < 5 h) time-series measured at Freshwater Canal Locks during Harvey (2017), Humberto (2007), and Gustav (2008); at Trident Pier during Irma (2017); at Naples during Hermine (2016) and Emily (2018), and at Clearwater Beach during Michael (2018) and Dennis (2005). The x-axis indicates the time in UTC, Day/Month/Year. Blue lines represent water level anomalies and correspond to the y-axis on the left; orange lines represent the lower frequency oscillations (tide + surge) and correspond to the y-axis on the right. Main meteotsunamis are indicated with gray shading.

maximum surge. Oscillations in the meteotsunami band persisted for hours, until the peak of the storm hit Trident Pier. Tropical storm Emily (2017) and Hurricane Hermine (2016) had maximum meteotsunami elevations of 0.45 and 0.43 m, respectively, measured at Naples (Florida). Hurricane Michael (2018) triggered maximum meteotsunami elevations of 0.42 m at Clearwater Beach (Florida). In these three events, single-peak meteotsunami waves were observed. Dennis (2005) produced maximum meteotsunami elevations at Clearwater Beach, with a single-peak wave of 0.41 m, followed by lower intensity trains of waves ahead of the maximum surge.

Most of the NOAA tide gauges along the Gulf of Mexico and Eastern United States are located inside estuaries or in the intra-coastal waterway, sheltered from the direct impact of open ocean waves. Only a few tide gauges are located in open waters, such as

at Naples and Clearwater Beach. These are shallow tide gauges, with mean water depths of ~1 m. Freshwater Canal Locks and Trident Pier are deeper gauges, with mean water depths of ~6 m, placed in channels connected to the open ocean.

Even though most of the extreme events had a clear maximum at a particular station, they also produced lesser maxima at nearby open ocean gauges. For example, during Hurricane Harvey (2017), water level oscillations in the meteotsunami frequency band extended from southern Texas to the Louisiana coast before and after Harvey's first landfall (Fig. 2). The intensity of these oscillations varied regionally, with minimum values (~0.20 m) at Galveston Bay Entrance (Texas) and the highest values (~0.78 m) at Freshwater Canal Locks. Water level observations collected from an open ocean tide gauge at Bob Hall Pier (located on the left of the track of Harvey) indicated the presence of

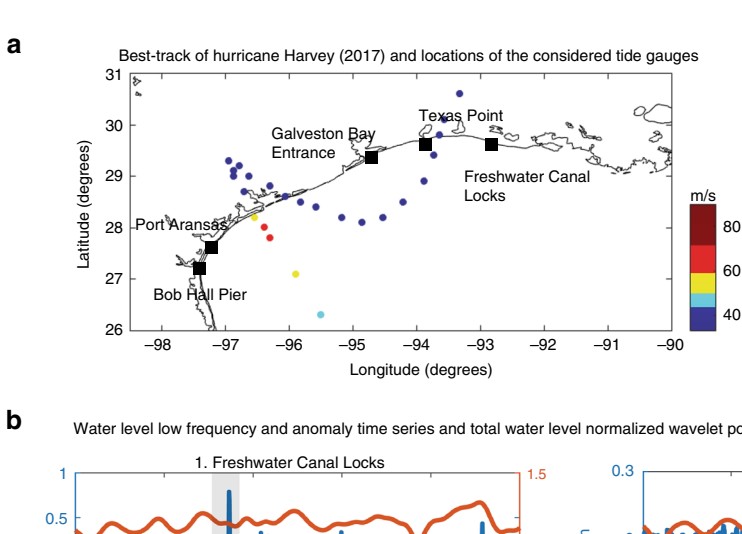

**Fig. 2 Water levels during Harvey (2017). Water level anomaly time-series and normalized wavelet power spectrum during Harvey (2017). a** Locations of the considered tide gauges (squares), the best-track (dots), and maximum sustained wind speeds (dot color) of Harvey (2017); **b** Water level time-series and total water level normalized wavelet power spectrum in the stations most affected by Hurricane Harvey (2017). Water level anomalies are indicated by blue lines and the corresponding y-axis is on the left; variations due to tides and surge are represented by the orange lines and the corresponding y-axis is on the right. The x-axis indicates the time in UTC, Month/Day. The contour lines in the normalized wavelet power spectrum indicate the 95% confidence level against red noise. Translucent portions of the diagrams represent data outside the cone of influence. Main meteotsunamis are indicated with gray shading. Note: There is a difference in the scale of the water level anomaly between Freshwater Canal Locks and the rest of the stations considered.

meteotsunamis during the surge forerunner, a day in advance of the landfall of the hurricane. The train of meteotsunami waves persisted for almost the entire day. The initial meteotsunami waves had wave heights (distance from trough to crest) lower than 0.25 m, which then increased to 0.35 m during the peak of the surge. In those tide gauges located to the right of the track (at Galveston Bay Entrance, Texas Point, and Freshwater Canal Locks), the meteotsunami behavior was different: separate meteotsunami peaks occurred before and after landfall. At Texas Point, we were able to identify four meteotsunami wave pulses with a clear diurnal cycle between the 25 and the 28 of August. These oscillations are better visualized through the normalized wavelet power spectrum, which represents the temporal change of the normalized water level variance contained at different periods. The variability within the thick black contour lines is considered statistically significant with a confidence level of 95%.

From the analysis of all the meteotsunamis occurring during TCs, we were able to distinguish two types of meteotsunami events: a main single-peak wave (e.g., Harvey 2017, Emily 2016, Hermine 2016, and Michael 2018) and longer lasting (12–24 h) trains of waves (e.g., Humberto 2007, Gustav 2008, Irma 2017). A combination of the previous two types with a main single-peak wave followed by a longer lasting train of waves was also observed (Dennis 2005).

One question that arises from the present analysis is whether TC-induced meteotsunamis are comparable in magnitude to the most extreme meteotsunamis recorded in each area. The cumulative distribution of the maximum meteotsunami elevation represents the probability of the maximum elevation being lower or equal to a given value for a given meteotsunami. Considering all the meteotsunami events between years 1998 and 2018, we reconstructed the cumulative distribution of the maximum meteotsunami elevation in those tide gauges where significant TC-induced meteotsunamis were observed (Fig. 3). At Freshwater Canal Locks, there is a higher probability of larger meteotsunamis, whereas Clearwater Beach is the tide gauge with the lowest

probability. Although TC-generated meteotsunamis are not as frequent as winter meteotsunamis at these locations, the highest TC-generated meteotsunamis correspond to the upper region (>98.5 quantile) of the cumulative distribution, with the exception of those observed during Humberto (2007) and Gustav (2008), in which the maximum elevations were higher than the 80% quantile of the cumulative distribution. These results show that TCs are significant contributors of extreme meteotsunamis at these particular tide gauges.

**Meteorological observations during TC-induced meteotsunamis.** The analysis of the NEXRAD atmospheric reflectivity mosaics has revealed that seven of the eight (7/8) highest meteotsunami events were concurrent with the passage of TCRs (Fig. 4).

During Harvey (2017), between the 24 and 29 August, several TCRs propagated from Galveston Bay Entrance to Freshwater Canal Locks, with most of them traveling parallel to the coast. The TCR associated with the maximum meteotsunami propagated at the speed of 14 m s$^{-1}$, perpendicular to the local coastline. This specific TCR was propagating northeastward to Freshwater Canal Locks at 1800 UTC August 25 (see black ellipse and arrow in Fig. 4a). A surface cold pool with an air temperature drop of 3.6 °C and sustained wind speed and direction variations of 6.1 m s$^{-1}$ and 77°, respectively were observed within the passage of this specific TCR (Fig. 5a).

In the case of Humberto (2007), trains of spiral rainbands reaching the coast at Freshwater Canal Locks produced ocean wave pulses with four main peaks above 0.5 m. Each of those water level anomaly peaks was concurrent with the passage of TCRs (Fig. 4b, c). Before landfall, TCRs propagated perpendicular to the coast offshore from Freshwater Canal Locks with a mean translation speed of 14 m s$^{-1}$ (Fig. 4b). After landfall, TCRs propagated parallel to the coast with a translation speed of 10 m s$^{-1}$ (Fig. 4c). For this specific event and for Gustav (2008), atmospheric pressure, wind,

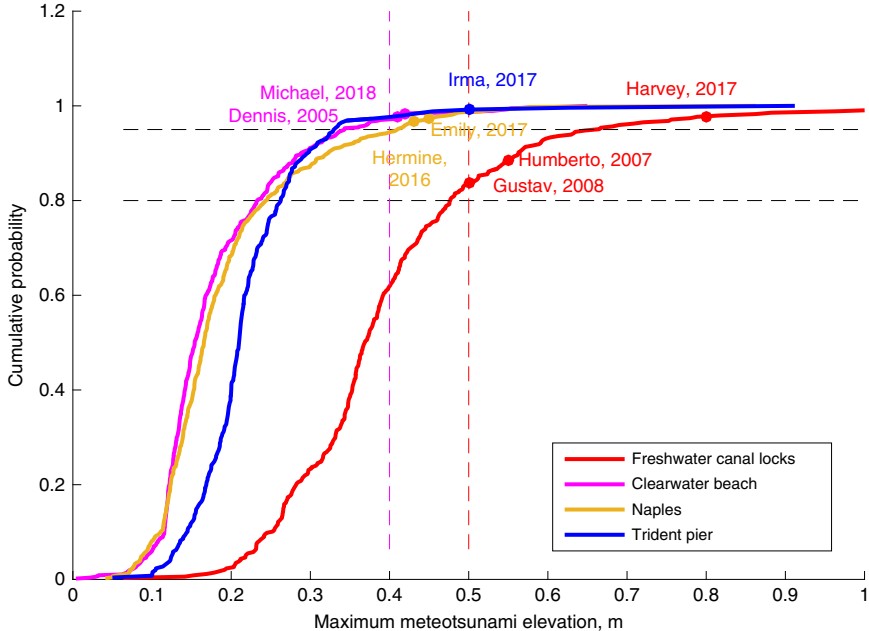

**Fig. 3 Cumulative distribution of the maximum meteotsunami elevations.** Cumulative distributions (lines) of the maximum meteotsunami elevations at the tide gauges where the extreme tropical cyclone (TC)-triggered meteotsunamis were observed: Freshwater Canal Locks (red), Naples (gold), Trident Pier (blue) and Clearwater Beach (magenta). The cumulative distributions were computed considering all the meteotsunami events between years 1998 and 2018. The maximum meteotsunami elevations of the most eight extreme TC-induced meteotsunamis are indicated with dots. The intersection of the cumulative distributions with the discontinuous gray lines represent the 80% and 98.5% quantiles.

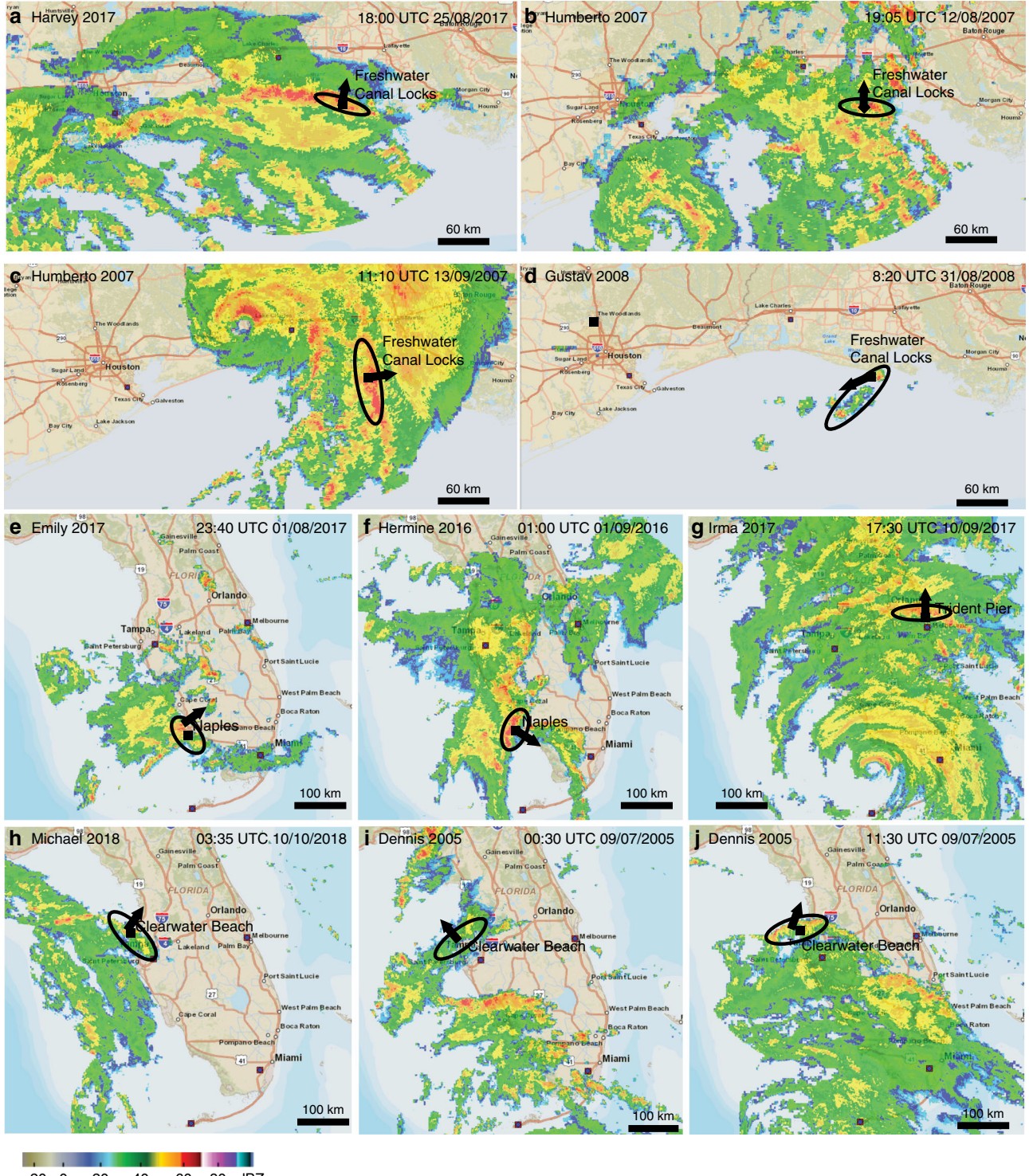

**Fig. 4 NEXRAD observations during tropical cyclone-induced meteotsunamis.** Atmospheric radar reflectivity mosaics during the eight most extreme meteotsunami events from the 1998–2018 Atlantic hurricane seasons. In each panel, the rainband that produced the main meteotsunami is shown within the black ellipse and the direction of propagation is represented with the black arrow. Dates are indicated as Day/Month/Year. dBZ decibel relative to Z (logarithmic dimensionless unit of radar reflectivity).

and air temperature observations are not available at the location of this tide gauge. During Gustav (2008), a squall propagated offshore Freshwater Canal Locks with a mean translation speed of 9.6 m s$^{-1}$ towards the southwest, parallel to the coast (Fig. 4d). This squall originated in the area of the Mississippi Delta and might not be directly associated with Gustav.

Water level oscillations in the meteotsunami band were strongest at Trident Pier during Hurricane Irma (2017). These were observed together with the passage of the spiral rainbands of Irma (outer rainbands followed by inner rainbands) (Fig. 4g). Atmospheric pressure oscillations with periods less than 2 h were recorded at the same station before and during the peak of the

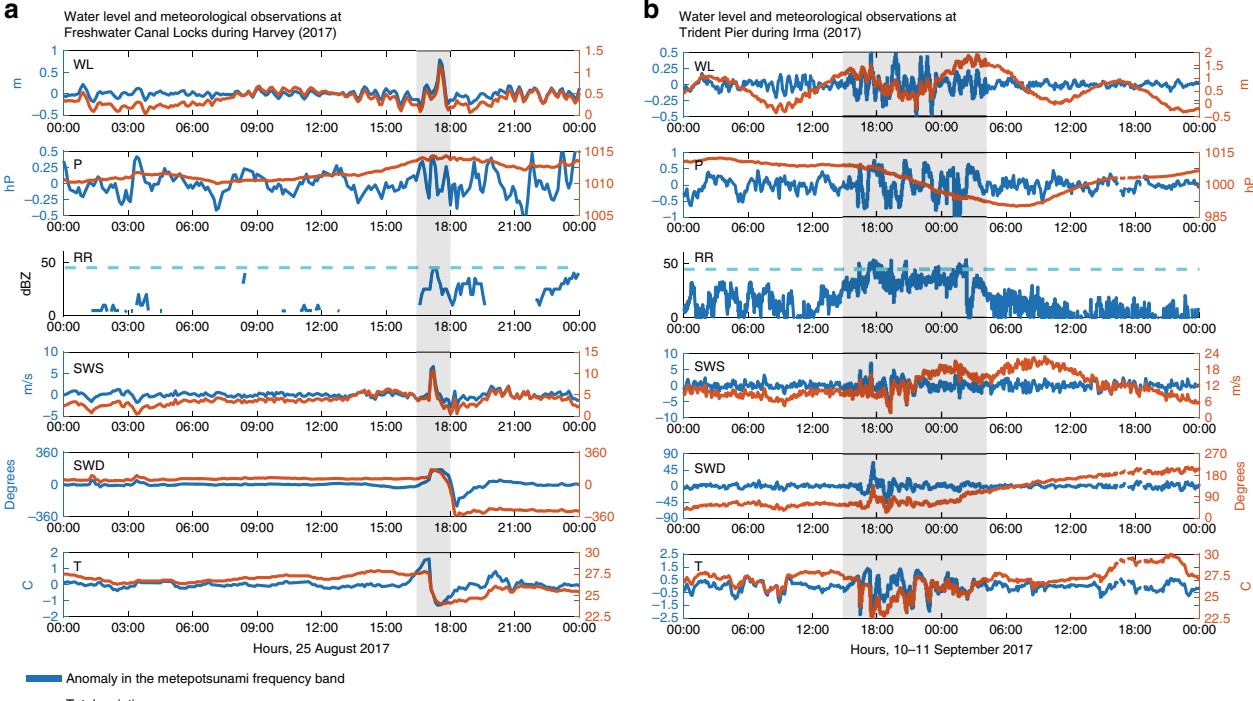

**Fig. 5 Meteorological observations during tropical cyclone-induced meteotsunamis.** Time-series of water level, sea level atmospheric pressure, atmospheric base radar reflectivity, sustained wind speed, sustained wind direction, and air temperature at **a** Freshwater Canal Locks during hurricane Harvey (2017) and **b** Trident Pier during Hurricane Irma (2017). High-frequency anomalies are indicated in blue and correspond to the y-axis on the left; total variations are represented with the orange lines and correspond to the y-axis on the right. The x-axis indicates the time in hours, Hour/Minute. The cyan line in the atmospheric base radar reflectivity panels indicate the 45 dBZ threshold usually used to define rainbands. WL water level (m), P sea level atmospheric pressure (hPa = hectopascal), RR atmospheric radar reflectivity (dBZ decibel relative to Z, logarithmic dimensionless unit of radar reflectivity), SWS sustain wind speed (m s⁻¹), SWD sustain wind direction (degrees), T Air temperature (°C).

storm (Fig. 5b). These were linked to sudden oscillations in the wind speed of up to 7 m s⁻¹, wind direction changes of up to 80°, and an air temperature drop of 2.9 °C (Fig. 5b).

During Emily (2017) and Hermine (2016) meteorological observations from the locations of the analyzed tide gauges and nearby meteorological stations were missing. Therefore, we supplemented the observations with sustained wind speed and direction and the air temperature measurements (30 min temporal resolution) from buoys from the University of South Florida. In the case of Emily (2017), we analyzed the measurements of the buoy located 130 km offshore from Naples at 50 m water depth (Station 42023). Wind speed anomalies of 5.8 m s⁻¹, wind direction changes of 57° and an air temperature drop 3.5 °C were observed 3 h before the meteotsunami hit Naples. The meteotsunami was concurrent with the passage of a rainband traveling normal to the coast (Fig. 4e) at a mean speed of 16 m s⁻¹. During Hermine (2016), we used the wave buoy located 160 km northwest of Naples, at 25 m water depth (Station 42013). Just before September 1, wind speed oscillations of up to 7 m s⁻¹, with direction changes of up to 109°, were recorded at the wave buoy. These fluctuations were linked to an air temperature drop of 2.7 °C. The arrival of the meteotsunami at Naples happened during the passage of a distant rainband traveling parallel to shore (Fig. 4f), with a translation speed of 9 m s⁻¹.

The arrival of the main meteotsunami during Michael (2018) also happened during the passage of a TCR that propagated parallel to the coastline (Fig. 4h) at a speed of 13 m s⁻¹. Associated with this rainband, 0.8 hPa atmospheric pressure fluctuations, 5.5 m s⁻¹ sustained wind variations and 2.2 °C air temperature changes in the meteotsunami frequency band were detected. Similar radar reflectivity conditions (Fig. 4i) were observed at the same meteorological station during the main

meteotsunami peak associated with Dennis (2005). A rainband propagated towards the northeast with a translation speed of 8.3 m s⁻¹. The main meteotsunami peak was followed by a sequence of trains of waves that lasted until the peak of the storm reached Clearwater Beach. These oscillations were associated with distant rainbands propagating northeastward, as observed from the radar reflectivity mosaics. Although the rainband that created the main meteotsunami peak was normal to the coast at Clearwater Beach, it propagated parallel to the coast from south of Tampa Bay to St. Petersburg. Meteorological observations during Dennis (2005) have 1 h temporal resolution, which is not enough to extract the wind, atmospheric pressure and air temperature fluctuations associated with these rainbands.

The characteristics of the observed main meteotsunamis and of the associated meteorological perturbations are summarized in Table 1, from which we can conclude that these extreme meteotsunami events were associated with the passage of squall-line-like features characterized by convective precipitation, ocean surface cold pools, and wind convergence zones at low levels in the atmosphere (observed as changes in wind speed and direction).

**TC-induced meteotsunami effects on total water levels.** The severity of meteotsunami impacts depends on the maximum meteotsunami elevation, duration and the arrival time with respect to combined tide and surge. To analyze the relative effect of TC-induced meteotsunamis on total water levels, we applied the methodology used by Ozsoy et al.[14]. We compared the maximum meteotsunami elevation, the astronomic tidal level, and the surge at the moment of the maximum meteotsunami during the eight most extreme meteotsunami events (see Table 2). We also

**Table 1 Tropical cyclone-induced meteotsunamis and associated atmospheric conditions.**

|  | Harvey | Humberto | Irma | Gustav | Emily | Hermine | Michael | Dennis |
|---|---|---|---|---|---|---|---|---|
| ME (m) | 0.78 | 0.55 | 0.50 | 0.49 | 0.45 | 0.43 | 0.42 | 0.41 |
| MT (minutes) | 36 | 36 | 48 | 42 | 96 | 42 | 24 | 90 |
| Type | SP | Train | Train | Train | SP | SP | SP | SP+train |
| Pa (hPa) | 0.5 | NA | 1.5 | NA | NA | NA | 0.8 | NA |
| SWSa (m s$^{-1}$) | 6.1 | NA | 7 | NA | 5.8 | 7 | 5.5 | NA |
| SWDa (°) | 77 | NA | 80 | NA | 57 | 109 | 10 | NA |
| Ta (°C) | 3.6 | NA | 2.9 | NA | 3.5 | 2.7 | 2.2 | NA |
| TCRd(°) | Normal | Normal and parallel | Parallel | Parallel | Normal | Parallel | Parallel | Normal |
| TCRs (m s$^{-1}$) | 14 | 10–14 | 8 | 9.6 | 16 | 9 | 13 | 8.3 |

Summary of the characteristics of the eight most extreme meteotsunami events and characteristics of the atmospheric perturbations and rainbands associated with them.
ME maximum meteotsunami elevation, MT meteotsunami period in minutes (in the case of a train of waves the period is computed with the highest peak), Type single-peak meteotsunami (SP) or train of meteotsunami waves, Pa atmospheric pressure anomaly (hPa); SWSa sustain wind speed anomaly (m s$^{-1}$), SWDa sustain wind direction anomaly (m s$^{-1}$), Ta Air temperature anomaly (°C), TCRd direction of the TCR with respect to the local shoreline (parallel or normal); TCRs propagation speed of the TCR (m s$^{-1}$), NA not available.

**Table 2 Tropical cyclone-induced meteotsunami effects on total water levels.**

|  | Maximum meteotsunami level (m) | Tide (m) | Surge (m) | Total water level (m) | Meteotsunami elevation/total water level × 100 | Phase-lag with respect to max surge | Maximum surge TC (m) | Maximum meteotsunami level/Maximum surge TC × 100 |
|---|---|---|---|---|---|---|---|---|
| Harvey | 0.78 | −0.07 | 0.43 | 1.15 | 69 | −0.7 | 1.50 | 53 |
| Humberto | 0.55 | 0.23 | 0.16 | 0.94 | 58 | 15.7 | 0.59 | 93 |
| Irma | 0.50 | 0.47 | 0.51 | 1.49 | 34 | 7.9 | 2.34 | 21 |
| Gustav | 0.49 | 0.27 | −0.05 | 0.71 | 70 | 71.5 | 2.89 | 17 |
| Emily | 0.45 | 0.04 | 0.26 | 0.75 | 61 | −30.9 | 0.52 | 86 |
| Hermine | 0.43 | −0.12 | 0.35 | 0.66 | 65 | 20.6 | 2.23 | 19 |
| Michael | 0.42 | 0.47 | 0.60 | 1.49 | 28 | 6.5 | 2.51 | 17 |
| Dennis | 0.41 | −0.49 | 0.01 | −0.07 | 590 | 35.8 | 2.07 | 20 |

Meteorological observations during TC-induced meteotsunamis. Time-series of water level, sea level atmospheric pressure, atmospheric base radar reflectivity, sustained wind speed, sustained wind direction, and air temperature at (a) Freshwater Canal Locks during Hurricane Harvey (2017) and (b) at Trident Pier during Hurricane Irma (2017). The phase-lag between the maximum meteotsunami and the maximum surge is computed as the time difference in hours between the moment of the maximum surge and the moment of the maximum meteotsunami; positive values indicate that the maximum meteotsunami occurred ahead of the maximum surge.

computed the time difference between the meteotsunami arrival and the maximum surge, and the ratio between the maximum meteotsunami elevation and the maximum surge during the TCs.

In the eight most extreme meteotsunami events analyzed in this study, the meteotsunami wave peaks occurred at different tidal phases (Table 2): during high tide in Humberto (2007), Irma (2017), Gustav (2008), and Michael (2018); during low tide in Dennis (2005) and Hermine (2016); during mid-tide in Harvey (2017) and Emily (2017). Most of the meteotsunamis described herein occurred ahead of the maximum surge (Table 2, column 6), except during Emily (2017) and Harvey (2017).

Maximum meteotsunami elevations contributed more than 25% to the total water levels at the moment of the meteotsunami in all the cases (Table 2, column 5). For example, during Michael (2018), a maximum total water elevation of 1.49 m was measured at Clearwater Beach. The water elevations due to astronomic tides, storm surge and the meteotsunami wave were 0.47, 0.60, and 0.42 m, respectively. Therefore, the contribution of the meteotsunami to the total water level was ~28%. During Harvey (2017) and Michael (2018) the maximum total water level and the maximum meteotsunami elevation occurred concurrently, meaning that meteotsunami waves significantly contributed to the total water levels at these particular stations, with flooding reported at the entrance of Tampa Bay during Michael (2018). The maximum surge at Naples during Emily (2017) was 0.35 m, lower than the maximum meteotsunami elevation. During Hermine (2016), the maximum surge at Naples was 0.45 m, similar to the maximum meteotsunami elevation. Coastal

flooding was reported in Naples during both Hermine (2016) and Emily (2017).

The maximum surge associated with each TC occurred close to landfall. The last column of Table 2 includes the ratio between the maximum surge observed during each TC and the maximum meteotsunami elevation, which provides an estimate of the relative relevance of the meteotsunamis compared to the surge produced by the overall TC structure. During Harvey (2017), the maximum meteotsunami elevation at Freshwater Canal Locks was 0.78 m (with a maximum water level of 1.15 m); maximum water elevations and surge levels close to the landfall were observed at Port Aransas (Texas), with values of 1.70 m and 1.50 m, respectively.

**Meteotsunami generation and propagation.** To further verify that TCs can create the two types of meteotsunami events identified from the observations (one main single-peak wave, longer lasting trains of waves, and their combination), we applied the Coupled Ocean–Atmosphere–Waves–Sediment Transport (COAWST) modeling system[15] to an idealized TC propagating over an alongshore-uniform ocean with simplified bathymetry. The initial TC imposed in the simulation evolved into a Category 3 hurricane after 24 h. As the TC moved northwestward with a mean translation speed of 8 m s$^{-1}$, several spiral rainbands formed along with strong wind and pressure disturbances (up to 20 m s$^{-1}$ and 3 hPa, not shown). Snapshots of the modeled atmospheric radar reflectivity show the spiral rainband structure,

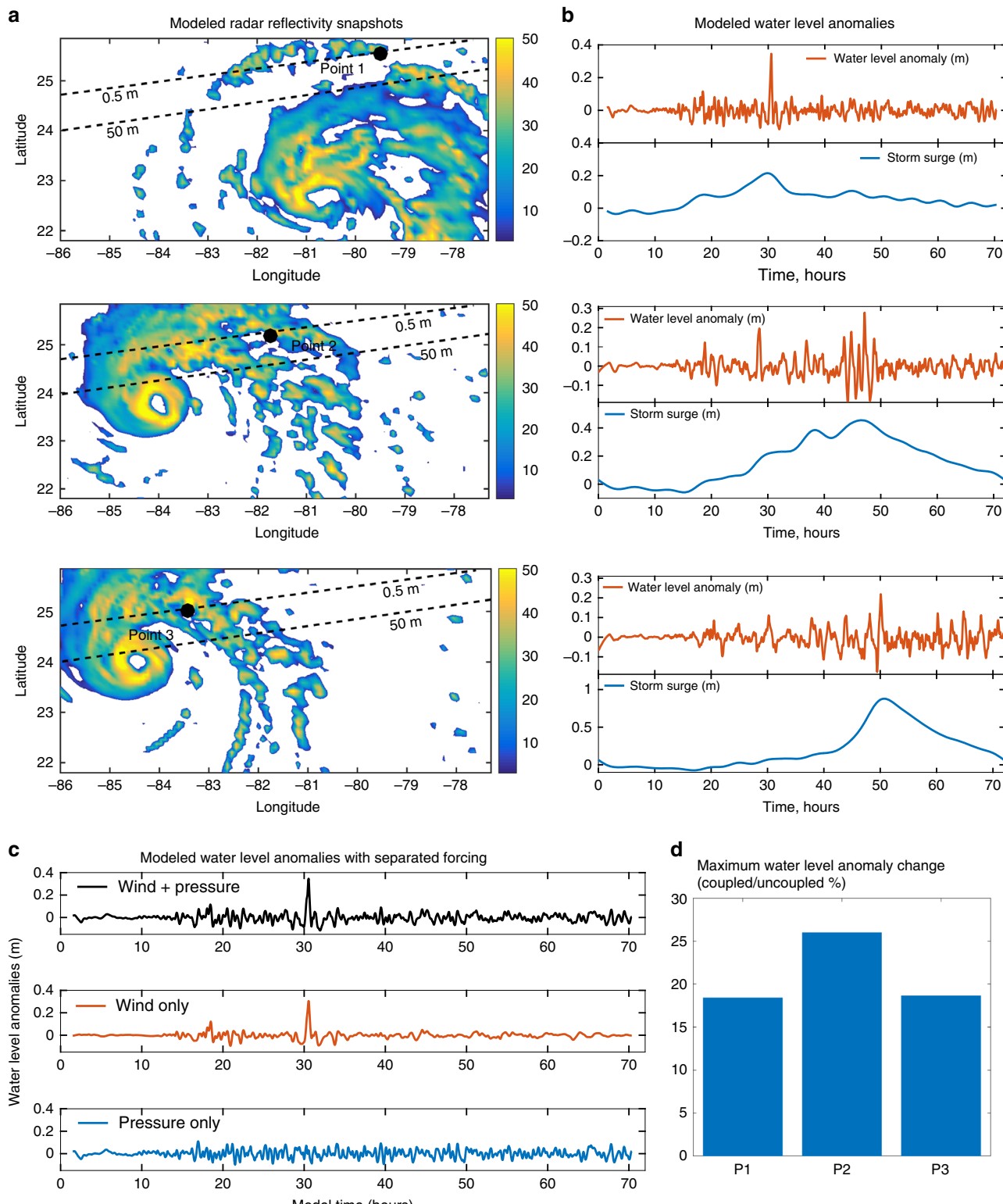

**Fig. 6 Meteotsunami waves triggered by an idealized tropical cyclone. a** Modeled atmospheric radar reflectivity snapshots of the idealized tropical cyclone, where radar reflectivity is given in dBZ (decibel relative to Z); the thick dash lines represent the 0.5 and 50 m isobaths; the figure also depicts the location of the coastal points (Point 1, Point 2, and Point 3) used to analyze the water level anomalies. **b** Time-series of modeled water levels at Point 1, Point 2, and Point 3; **c** Modeled water level anomalies with wind and pressure forcing (upper panel), with only wind forcing (middle panel) and with only pressure forcing (bottom panel); **d** Effects of wave-dependent ocean surface roughness indicated by maximum change (in percentage) of water level anomaly between coupled and uncoupled model applications.

with the characteristic inner and outer distant spiral rainbands (Fig. 6a).

These convective bands propagated with the TC and triggered different types of meteotsunamis along the coast (Fig. 6b). A main single-peak meteotsunami with maximum water level anomaly of 0.36 m was generated at Point 1 (upper panel in Fig. 6a) by a propagating distant rainband, 30 h after the simulation started. It happened at the same time as the local maximum surge (0.21 m) and contributed 63% to the total water level. Trains of meteotsunami waves were detected at Point 2 and Point 3. These meteotsunami waves were triggered by the inner rainbands (middle and lower panel in Fig. 6a) and showed smaller amplitudes compared to the single-peak wave. At Point 2, we can find a single-peak wave 28 h after the simulation started, followed by trains of waves at 42–48 h with a maximum water level anomaly of 0.29 m, which contributed 38% to the total water level. At Point 3, long-lasting trains of meteotsunami waves were observed both before and after the peak surge. The maximum meteotsunami wave happened at the peak of the storm surge and represented 20% of the total water level. The modeled types of meteotsunami waves showed similar spatial scales to propagating rainbands (not shown), with periods ranging from 1–2 h.

In our model application, we included wave coupling to better represent the wave effects on the sea surface roughness[16]. Compared with simulations without wave coupling, the wave-dependent surface roughness resulted in a maximum 18.4, 26.0, and 18.6% increase of maximum meteotsunami elevations at points 1, 2, and 3, respectively (Fig. 6d). To analyze the generation and propagation mechanisms of the TC-induced meteotsunamis, we first compared the water level anomalies under different driving forces: with wind and atmospheric pressure, just with wind and just with atmospheric pressure (Fig. 6c). With only pressure forcing, water level response generally follows the inverse barometer effect. Similar results apply to the trains of meteotsunami waves (not shown). Results indicate that wind forcing is the main trigger of the modeled meteotsunamis with >90% contribution to the maximum meteotsunami elevation.

Furthermore, we analyzed the meteotsunami generation evolution and propagation in the specific meteotsunami forced by the main outer spiral rainband (single-peak meteotsunami observed at Point 1). This TCR propagated normal to the coast. The analysis consisted in tracing the meteotsunami wave rays (see Methods) and quantifying the generation and propagation coefficients along the wave rays. Fig. 7a shows the wave rays traced along the meteotsunami path and their corresponding water level anomalies. From the 12 wave rays shown in Fig. 7a, we selected the three with largest water level anomalies.

From the initial points, propagation, and amplification of the meteotsunami waves depend on the wave shoaling and refraction, resonance between atmospheric forcing and ocean surface waves (Proudman resonance), and on the intensity of the atmospheric forcing. By comparing the evolution, propagation and generation coefficients ($K_e$, $K_p$, and $K_g$, described in the Methods section), Fig. 7b indicates that, in the region with water depths < 15 m, wave shoaling was the dominant process for meteotsunami amplification. However, in the offshore region (water depths > 15 m), wind stress was dominant. As expected, wave refraction showed small effects on the meteotsunami propagation in this normal incidence case. In addition, to isolate the effect of Proudman resonance, we calculated the efficiency of the energy transfer from the atmosphere to the ocean, which is given by ratio between generation coefficient ($K_g$) and wind divergence divided by water depth (Fig. 7c). This coefficient increases from 45 m water depths to maximum values at water depths between 15–20 m; this specific rainband propagated with a mean translation speed of 13 m s$^{-1}$, with a Proudman resonance depth of 17 m. At

shallower water depths, the energy transfer efficiency decreased, the meteotsunami wave started to decouple from the atmospheric disturbance and wave shoaling became the dominant amplification mechanism (Fig. 7b).

Since the generation and amplification of meteotsunamis might be sensitive to the translation speed and propagation direction of the rainbands, we compared the modeled maximum meteotsunami elevations forced with varied rainband propagating speed (ranging from 7.5 m s$^{-1}$ to 25 m s$^{-1}$) and direction (ranging from 42° to 118° with respect to the coastline). Fig. 7d shows that modeled maximum water level anomalies exceeded 0.3 m with rainbands propagating at 11 m s$^{-1}$ and 15 m s$^{-1}$; either faster or slower translation speeds reduced the meteotsunami amplification. Modeled maximum meteotsunami elevations were maximum in the case of 90°, which is the case when the rainband was propagating perpendicular to the coast. Rainbands with oblique incidence reduced the maximum meteotsunami elevation.

## Discussion

The potential of TCs to trigger meteotsunamis was demonstrated by Mercer et al.[5], who analyzed how tropical storms Jose (1999) and Helene (2000) created barotropic waves with periods of ~10 min across the Grand Banks of Newfoundland. The low-pressure atmospheric systems associated with these two tropical storms propagated at a translation speed of ~30 m s$^{-1}$, which created Proudman resonance over the shelf and forced wave wakes in the rear side of the storms. Unlike the barotropic waves described by Mercer et al.[5], the meteotsunamis identified and analyzed in this study are not associated to the whole structure of the storm but rather with atmospheric features with much smaller scales: the spiral rainbands (TCRs).

Our analysis of water level observations during the 1998–2018 Atlantic hurricane seasons demonstrates for the first time that meteotsunamis are frequently triggered by TCRs. In more than half the events, the maximum meteotsunami elevations were higher than 0.2 m, in 20% of the cases greater than 0.3 m, and in 8% of the events higher than 0.4 m. We were able to distinguish two different types of meteotsunamis: single-peak waves and trains of meteotsunami waves. Persistent meteotsunami wave trains with durations of ~12 h are more likely to occur in the front of the TC, such as observed at Bob Hall Pier during Harvey (2017) and at Trident Pier during Irma (2017). The minimum distance between the eye of Harvey (2017) and Bob Hall Pier was 60 km, and Trident Pier was located at a minimum distance of 170 km with respect to the eye of Irma (2017). Single-wave type meteotsunamis were observed at larger distances from the eye of the TC (e.g., ~450 km during Harvey (2017), ~350 km during Michael (2018)). During Harvey (2017), the observed meteotsunami trains that propagated from Galveston Bay Entrance to Freshwater Canal Locks were coincident with the passage of several spiral rainbands associated with Harvey (2017). We have shown observational evidence that meteotsunami behavior differs according to where they are generated—they tend to manifest as persistent trains of waves when closer to the eye of the TC, whereas distant meteotsunamis are more likely to emerge as single-peak meteotsunami waves.

In the eight most intense meteotsunami events identified from the 1998–2018 Atlantic hurricane seasons, maximum meteotsunami elevations were coincident with atmospheric pressure, wind (sustained and gust), and cold air temperature fluctuations with periods less than 2 h. The analysis of atmospheric NEXRAD reflectivity mosaics from NOAA showed that in all analyzed cases, the arrival of the meteotsunami was concurrent with high atmospheric radar reflectivity (>45 dBZ), which is indicative of the passage of rainbands. The observed link between water level

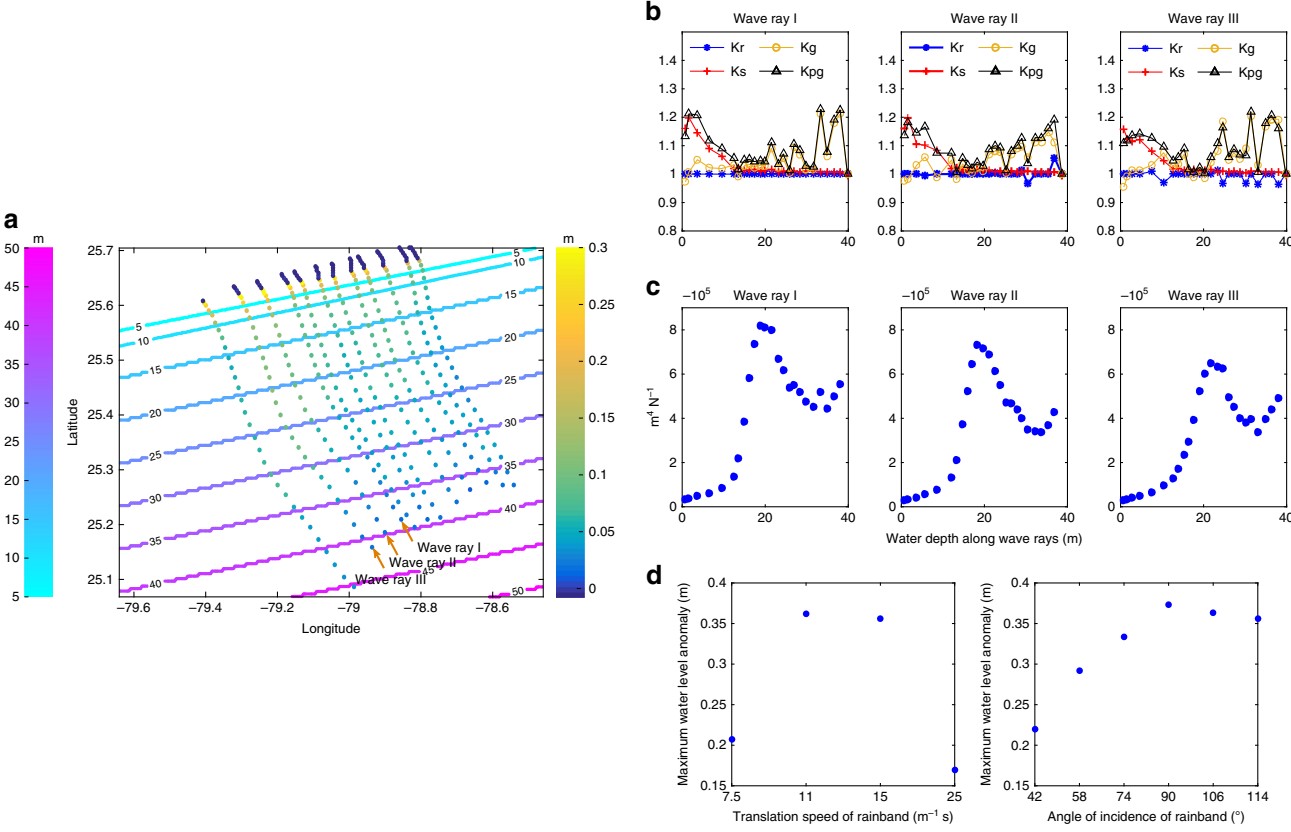

**Fig. 7 Meteotsunami generation and propagation in the idealized tropical cyclone. a** Considered wave rays. The idealized bathymetry is represented with the colored lines (color map on the left), modeled water level anomalies along the wave ways are depicted with colored dots (color map on the right); **b** Generation, evolution, and propagation (shoaling and refraction) coefficients along the selected wave rays; the y-axis represents the dimensionless evolution, propagation, refraction, and shoaling coefficients; **c** Ratio between the generation coefficient and wind divergence divided by the mean water depth ($\Gamma$, $m^4 N^{-1}$) along selected wave rays; **d** Effects of the translation speed and angle of incidence of the rainband on the maximum water level anomaly.

anomalies, atmospheric pressure, wind, air temperature, and atmospheric radar reflectivity suggests that TCRs with squall-line-like structure are the main trigger of these types of ocean waves. The fact that the most energetic meteotsunami waves are observed simultaneously with atmospheric fluctuations suggests that these are forced ocean waves.

The analysis of water levels during the most extreme TC-triggered meteotsunamis has revealed that these waves can significantly contribute to the total water levels and enhance flooding probability in areas distant to landfall where maximum surges are expected. For example, during Emily (2017) and Hermine (2016), maximum water elevations occurred during the passage of TC-induced single-peak meteotsunami waves, which significantly contributed to the total water levels; flooding occurred at Naples as a result. Longer lasting meteotsunami wave trains usually travel ahead of the peak of the storm and can continue until the maximum surge. The relative magnitude of these meteotsunamis with respect to the maximum surge is expected to be smaller, but they still can contribute to the increased water levels before and during the peak of the storm. Although not included in the present analysis, preliminary observations during Hurricane Dorian (2019) also show meteotsunamis along the coast of North Carolina. These had maximum elevations of 0.2 m at Duck (North Carolina) and they were concurrent with the maximum surge (0.95 m). Meteotsunamis occurring at the peak of the surge have also been identified in other TCs, such as during Isabel (2003) with maximum elevations of 0.2 m at Duck, and during Irene (2011) with maximum elevations of 0.27 m at Atlantic City (New Jersey).

We corroborated our observed meteotsunami behavior and the potential of TCs to trigger meteotsunamis by applying the COAWST modeling system to idealized TC cases. Once we verified that the model is able to reproduce the types of meteotsunamis identified in the observations, we used the idealized model to analyze the relative role of wind-stress, atmospheric pressure, and wind wave induced ocean roughness. Results showed that the wind-stress divergence is the key forcing for TC meteotsunami generation, unlike the winter meteotsunamis in the Gulf of Mexico or the meteotsunamis observed in the Mediterranean Sea in which the atmospheric pressure fluctuations play a primary role. In the idealized meteotsunami event analyzed herein, wind wave induced ocean surface roughness increased the magnitude of wind shear stress and this translated into an increase of the meteotsunami magnitude of about 18–26%.

An idealized model has been applied to analyze the meteotsunami propagation and generation processes. TC-generated meteotsunami waves are initially forced waves and travel with the atmospheric fluctuations that trigger them. The amplitude of the meteotsunami in each case depends on the strength of the atmospheric fluctuation, its own translation speed, the direction of propagation, and the bathymetric configuration. The transfer of energy from the atmosphere to the ocean becomes more effective as the speed of the atmospheric fluctuations and the shallow water wave celerity come closer into agreement. If the celerities of the atmospheric perturbation and ocean shallow water wave start to differ (e.g., due to bathymetric variations), meteotsunamis become free waves.

Meteotsunami generation is highly dependent on bathymetric characteristics. Seven of the eight meteotsunamis with the greatest intensities observed during the 1998–2018 Atlantic hurricane were measured by tide gauges along the Northern Gulf of Mexico. A priori, this could imply that the bathymetric characteristics of the Northern Gulf of Mexico are more appropriate for meteotsunami generation and propagation. However, this study cannot confirm such a hypothesis since the majority of the measured extreme TC events propagated along the Gulf of Mexico in our analysis sample. To explore this theory further, high spatial and temporal resolution measurements from TCs propagating from other areas would be required, which are rare and hard to obtain. Further studies focused on the effects of TC characteristics (intensity, translation speed, ambient oceanographic conditions) on the meteotsunamis generation potential are also needed.

The Gulf of Mexico and the Florida East Coast are micro-tidal with low-lying terrain. Any contribution to the total water levels increases the elevation at which gravity and infragravity waves will be acting. Meteotsunamis can increase the risk for flooding, dune erosion, and sandspit or barrier-island breaching in coastal regions, especially those close to the eye of the TC, where the surge elevation is larger. Meteotsunamis can also force strong rip-currents[17] and tsunami-like currents with large potential to transport sediments.

Because of the inherent ability of meteotsunamis to increase coastal damage and erosion, meteotsunami effects must be incorporated into future models for accurate flood-risk forecast and assessment. For example, the design of coastal structures and coastal-flood insurances are based on the total water levels estimated for a given return period. The total water levels for a given return period could be under-estimated if meteotsunamis are not considered. In regards to flood-risk forecasts during hurricane warnings, the consideration of meteotsunamis could improve the accuracy of the forecast. However, this would require the atmospheric models to accurately forecast specific TCRs and the hydrodynamic models to be forced with high temporal (~5 min) and spatial (~500 m) resolution wind and sea level atmospheric pressure fields. Despite considerable advancements in knowledge and modeling capabilities of the inner structure of TCs, forecasting specific TCRs is still challenging. Moreover, feeding the hydrodynamic models with higher resolution atmospheric fields could result in a prohibitive increase in computational cost. Further research on meteotsunami forecasting is needed to overcome these limitations.

## Methods

**Water level and atmospheric observations**. In this study, we define meteotsunamis by the temporal scale of the water level variations, following Monserrat et al.[8]. According to their definition, storm surges are the result of the overall atmospheric pressure and wind structure of a given storm and produce basin-scale impacts, whereas meteotsunamis are induced by the atmospheric features with shorter spatial and length scales and have more local effects.

For each TC considered in this study, we analyzed all the quality-controlled meteorological and water level observations from NOAA along the United States East Coast, Gulf of Mexico, and Puerto Rico. Water level observations have a temporal resolution of 6-min (1 min resolution measurements are available, but these are not manually quality-controlled). We processed these measurements by removing the tidal signal with the predicted tides from NOAA and using a Lanczos filter[18] with a cutoff period of 5 h. The Lanczos filter is a Fourier method of filtering digital data designed to reduce the amplitude of the Gibbs oscillation. We used a cutoff period of a time interval (5 h) larger than what is usually associated with meteotsunamis (2 h) to ensure that we captured all the possible oscillations in the meteotsunami frequency band. Water level anomalies are usually small, of the order of few centimeters. For example a 1 hPa variation in the sea level atmospheric pressure produces ~0.01 m variation in the water level due to the inverted barometer effect. The presence of meteotsunamis is defined as water level anomalies that exceed (in absolute value) six times the standard deviation of the water level anomaly. While this threshold is higher than the value suggested by Monserrat et al.[8], it is more conservative and appropriate for our specific study sites because it ensures the exclusion of waves triggered only by the inverted barometer effect[7]. For each TC, we computed the maximum water level

anomalies within the Gulf of Mexico, Eastern Unites States, and Puerto Rico. We used wavelet power spectrum analyses to show the temporal change of the variance contained at different periods. We applied this analysis to water level, sea level atmospheric pressure, sustained wind speed, sustained wind direction, and air temperature time-series. The normalized wavelet power-spectrum represents how the period distribution (y-axis) of the normalized variance of a given time-series changes in time (x-axis). The period-time domain at which the variability of the signal is significant is represented by the 95% confidence level against red noise, which is indicative of the statistical significance of the variance. Translucent portions of the diagrams represent the data outside the cone of influence (the region in which the edge effects become relevant). We used the Matlab toolbox by Grinsted et al.[19], to compute and plot normalized wavelet power spectra, using a Morlet wavelet (with a nondimensional frequency = 6) for the continuous wavelet transform. The cumulative distributions of the maximum meteotsunami elevation climatology were computed following the method proposed by Olabarrieta et al.[7], in which the Hilbert transform is used to estimate the envelope of the water level anomaly and from which each single meteotsunami event is extracted. We used the NEXRAD atmospheric radar reflectivity mosaics from NOAA to depict the spatial and temporal structure of the TC rainbands and to estimate their speed and direction of propagation.

**Coupled ocean–atmosphere–coupled idealized model**. We simulated meteotsunami generation and propagation processes using an idealized TC with the COAWST modeling system[15]. This open-source modeling system is comprised of several components, including the Weather Research and Forecast (WRF) model[20,21], the Regional Ocean Modeling System (ROMS)[22,23], and the Simulating Waves Nearshore model (SWAN)[24]. A coupler, the Model Coupling Toolkit (MCT)[25], provides the platform for data exchanges between model components. Several studies focused on the modeling meteotsunamis in the Balearic Islands[26], in the Adriatic Sea[27] and in the Northern Gulf of Mexico[28] have successfully used COAWST.

**Weather research and forecast model**. The WRF model[20,21], including the Advanced WRF (ARW) core, is a nonhydrostatic, fully compressible numerical model, widely used in mesoscale numerical weather predictions. It uses Arakawa-C grid and a terrain-following hydrostatic pressure coordinate system in the vertical direction, and incorporates multiple schemes for physical processes including microphysics, planetary boundary layer, and short and long wave radiations. The WRF atmospheric model has been proven to be skillful when simulating and forecasting hurricane intensity and evolution in numerous previous studies[20,29,30]. To produce and study TCs in controlled environments, we used the idealized TC modeling framework developed by Nolan[31]. This system allows the user to define the initial atmospheric sounding, wind profile, and the sea surface temperature of a large-scale environment, and the initial structure of a balanced vortex that will evolve over several days into a mature TC. This framework has been used in many studies focused on the effects of different thermodynamic environments[32,33], environmental wind profiles[34–36], and vortex size and translation speed[37] on TC structural evolution and intensity.

In this idealized simulation, the initial vortex configuration of WRF was set with a radius of maximum wind (RMW) of 72 km, a maximum wind velocity of 30 m s$^{-1}$, a modified Rankine vortex decay rate of 0.3, and meridionally varying surface temperature ranging from 25 °C to 30 °C. The background wind speed at the surface was set to −8 m s$^{-1}$ (moving westward). The initial atmospheric temperature and humidity profiles are based on the Dunion moist tropical sounding[38], with mid-level humidity reduced by 20% as in Nolan and McGauley[33]. The simulations used an outer domain of 651 × 651 grid points with 9 km grid spacing, and a single, nested, vortex-following grid of 450 × 450 points with 3 km spacing. We used the microphysics scheme of Thompson et al.[39]. Heat fluxes, sea surface atmospheric pressure, and winds at 10 m with a temporal resolution of 5 min were stored in both WRF grids and used to force the ocean and wave models. In this idealized case, WRF was run separately from ROMS and SWAN (no feedbacks from the ocean models were considered).

**The Regional Ocean Modeling System**. ROMS[22,23] is a three-dimensional hydrostatic ocean circulation model with terrain-following coordinates that solves the Primitive Equations based on the Boussinesq approximation. ROMS was configured in barotropic mode, disregarding water density variations. The horizontal grid was spherical with a resolution of 0.005 degrees (~500 m), and in the vertical five equally spaced layers were used. In this specific simulation, an alongshore-uniform bathymetry, simplified from real bathymetry in the Northern Gulf of Mexico, was considered. The deep ocean and the continental shelf have constant water depths of 1000 and 50 m respectively; from the continental shelf, the water depth decreased with a constant slope of 0.0005–0.5 m along the coastline. Initial conditions were set to zero for the velocities and free surface elevation. A Flather[40] boundary condition for the lateral and offshore barotropic velocities and a Chapman condition for the free surface elevation were imposed. ROMS was forced with the sea surface atmospheric pressure and the wind velocity at 10 m produced by WRF from days 1–3. The COARE 3.5 algorithm was used to compute the wind stress for ROMS. SWAN was run in the same grid as ROMS. To compute the wind shear stress, we used the Taylor and Yelland[16] closure model.

**Simulating Waves in the Nearshore model.** Simulating Waves in the Nearshore (SWAN)[24] is a third-generation wave model that solves the wave-action balance equation that describes the evolution of the action density in space and time. It incorporates several source and sink terms including: wind velocity input, bottom friction, quadruplet wave-wave interactions, white-capping, and depth-induced breaking in shallow waters. Statistical wave parameters, such as the significant wave height or peak period, are derived from the directional wave spectrum solved by the governing equation. In this application, the same horizontal grid as in ROMS is used. The frequency and directional domains are solved with 24 and 72 bins, respectively. In this case, wave generation and propagation within the computational domain is considered. The effects of ocean gravity waves on ROMS are included through the vortex-force method[41] and the increase of the ocean surface roughness. In this study, the first effect is disregarded, because our objective was to ascertain the potential of different atmospheric structures within TCs to trigger meteotsunamis, and the correct modeling of the surf zone currents produced by waves requires higher horizontal grid resolutions than those considered in this analysis.

**Meteotsunami generation and propagation.** We used a method to analyze the meteotsunami propagation and generation inspired by Sheremet et al.[42]. This method consists in the computation of the meteotsunami generation, evolution, and propagation coefficients along wave rays. To compute the wave rays we used a method that differs from Sheremet et al.[42]. In this specific case, we filtered the barotropic velocities and water elevations at the meteotsunami frequency band from the ROMS outputs. Once we identified the main rainband that created a single-peak meteotsunami wave at Point 1, we selected different points along the location of the rainband at a water depth around 45 m. We assumed that the linear wave theory can be used to approximate the phase celerity of these meteotsunami waves. We used the barotropic velocities along the wave crests to compute the direction of propagation of the meteotsunami crest. With this information, we followed the wave rays or the paths of several points along the wave crest from water depth 45 m to the coast. The total evolution coefficient ($K_e$) of the meteotsunami along the wave ray was computed as the ratio of the maximum meteotsunami elevation between consecutive points along a wave ray. The total evolution coefficient is defined as the product of the propagation coefficient ($K_p$) due to the combined effects of shoaling ($K_s$) and refraction ($K_r$), and the generation coefficient ($K_g$). We assumed that the effects due to bottom friction and diffraction can be neglected.

$$K_e = K_p \cdot K_g = K_r \cdot K_s \cdot K_g \tag{1}$$

With the bathymetric variations along each wave ray, and applying Green's Law, we computed the shoaling coefficient. The refraction coefficient was estimated with the variation in the distance between two adjacent wave rays. The generation coefficient represents the increase of the meteotsunami amplitude due to the transfer of energy from the atmosphere. This was estimated as the ratio between the evolution coefficient and the propagation coefficient along each wave ray.

$$K_g = K_e \cdot K_p^{-1} \tag{2}$$

The wind-stress divergence $\nabla \cdot \tau$ divided by the local water depth represents the atmospheric force the relative to water depth, $\Gamma = \frac{|\nabla \cdot \tau|}{h}$. The ratio between $K_g$ and $\Gamma$ is an indicator of the efficiency of the energy transfer between the atmosphere and the ocean.

To determine whether the meteotsunami is a bound or a free wave, we determined the timings of the meteotsunami crest and the maximum wind-stress divergence arrivals along the wave rays. If the meteotsunami is bounded to the atmospheric disturbance, the ocean wave and the atmospheric fluctuation travel at the same speed and occur concurrently at a given point along the wave ray. If the meteotsunami wave is a free wave, the atmospheric fluctuation and the meteotsunami are observed with a phase-lag.

## Data availability

Atmospheric radar reflectivity mosaics (http://gis.ncdc.noaa.gov/map/) and atmospheric and water level observations from NOAA Tides & Currents (https://tidesandcurrents.noaa.gov/) are used in this study. These observations are publicly available.

## Code availability

The Coupled Ocean–Atmosphere–Wave–Sediment Transport (COAWST) modeling system is an open-source code and can be downloaded from the USGS Github Code Repository: https://code.usgs.gov/coawstmodel/COAWST. We used the Matlab toolbox by Grinsted et al.[19]. (https://github.com/grinsted/wavelet-coherence) to compute and plot normalized wavelet power spectra. The Matlab scripts to analyze the time-series and for the statistical analysis are available from the authors. Atmospheric radar reflectivity mosaic maps in Fig. 4. were created with the NOAA iterative mapper (https://gis.ncdc.noaa.gov/maps/ncei/radar).

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

## Acknowledgements

We thank all the developers of COAWST, ROMS, WRF, and SWAN models. D.N. was supported by NSF grant AGS-1654831. We would like to thank Dr. K. Bagamian for her editorial and writing suggestions. We would like to thank Dr. A. Aretxabaleta for the internal US Geological Survey internal revision and suggestions.

## Author contributions

L.S. performed the idealized TC atmospheric simulations with the WRF model, performed the meteotsunami modeling with ROMS and SWAN, downloaded the radar reflectivity time-series, and assisted in the writing of the manuscript. M.O. developed the conceptual framework and methodology, conducted data analysis, and wrote the manuscript. D.S.N. provided the modeling framework for the idealized TC simulations, and assisted in the development of the simulations, the meteorological analysis, and the writing of the manuscript. J.C.W. assisted with the numerical modelling and data analysis. L.S., D.S.N. and J.C.W. reviewed and edited the manuscript.

## Competing interests

The authors declare no competing interests.
