## [Peer Review File · Nature Communications]

Reviewers' comments:

Reviewer #1 (Remarks to the Author):

The manuscript presents the presence of meteotsunamis triggered by landfalling hurricanes in observations and idealized simulations. This could be a significant piece of work. However, improvements are needed before it is considered acceptable in this highly visible journal.

(1) It seems that it is already generally known that TCs have the potential to generate meteotsunamis. I wonder if the findings of this work are novel or significant. The authors need to make the following points clear.

- a) Is it known previously that TCs can generate meteotsunamis larger than 0.4 m?
- b) Is it reported previously that two types of meteotsunamis exist under TCs?
- c) Is the magnitude of meteotsunamis reported in this study significant compared to those observed under other conditions?
- d) Only 4 out of 16 analyzed cases have meteotsunamis larger than 0.4 m. Please comment on if you think this is frequent.

(2) Lines 72-74 It seems this work did not make any attempt to answer these questions either.

(3) Lines 212-248 There are a lot of discussions based on the figures not shown in the main content. I suggest either greatly shortening this section or incorporating the related figures to the main content. Particularly, I suggest moving Table 2 in supplementary materials to the main content.

(4) The analysis of the idealized simulation is a bit cursory. The analysis should be expanded to better reveal the initiation, evolution and characteristics of the modeled meteotsunamis.

- a) Is wave-coupling necessary in this application?
- b) What are the spatial scale and horizontal structure of these waves?

Reviewer #2 (Remarks to the Author):

I found the manuscript very interesting and the topic deserves to be published in a high-rank journal. With an increased availability of high-resolution sea level data, at a minute sampling rate, the research on extreme sea levels found that meteotsunamis and nonseismic sea level oscillations at the tsunami timescale are contributing substantially to the extremes.

Yet, the manuscript lacks a number of important items and analyses, which should be part of it in the case of its publishing in Nature Communications. These are:

1. What is an overall intensity of TC-induced meteotsunamis compared to other types of meteotsunamis? Are TCs minor or major process driving meteotsunamis in the investigated areas? I want to see the analysis of these extreme TC-driven meteotsunami waves versus the climatology of meteotsunami waves at stations where the largest waves are recorded, of course for the period when the high-frequency sea level data is available (and that is at least 10-15 years, I think). By doing that, the overall intensity of TC-induced meteotsunami waves vs. meteotsunami waves induced by other atmospheric processes may be assessed and should be part of the article.
2. I don't see any discussion regarding bathymetry that is in front of tide gauges. Are tide gauges at the open sea or inside harbours? What is the depth in front of gauges? Is it favourable to generation of meteotsunamis (disturbance speed vs. long ocean waves speed  Proudman resonance?)? Also, I do not see the proof that meteotsunami wave was a solitary wave - this should be proven by mathematics, i.e. that nonlinear and dispersive effects are cancelled in such a wave, so that it maintain its shape over a path. I think that the authors used the wrong terminology here. It looks to me that single atmospheric disturbance is generating the meteotsunami wave though Proudman resonance, and the wind is the dominant mechanism over air pressure. All of these should be analysed and discussed in the text.
3. Above all of four quoted TC-generated meteotsunamis (Harvey, Irma, Hermine, Michael), the only TS moving parallel with the coast is Irma, during which the series of meteotsunami waves are generated only. Is there any connection between the pathway of a TC (parallel vs. perpendicular to the coast) and meteotsunami waves? I think yes, but the authors should investigate this. Hint: look the study by Sheremet et al. (2016, Natural Hazards, doi: 10.1007/s11069-016-2446-2). The authors may use numerical simulations for doing that.
4. The modelling exercise does not bring anything new to the research, as basically trying to reproduce air pressure and wind effects to the barotropic ocean. For that reason, I would omit it as aggravates the findings, or to use it to prove the underlying physics. As it stands now, it does not provide any new findings regarding physics and processes related to meteotsunami waves. Alternatively, the authors may dig deeply into the physics and go beyond is presented (e.g. following previous comments).
5. The detailed sections on specific hurricanes should be condensed in the manuscript, while details should be moved to supplementary material. Particularly if including new analyses as suggested above.

Reviewer #3 (Remarks to the Author):

This is an interesting paper that considers water level variations during tropical cyclones and argues that meteotsunamis are frequently developed. It provides good evidence that this is the case and relatively long-period waves of a few minutes to a few hours are observed. It is also shown that these waves are consistent with generation by pressure and wind fluctuations during tropical cyclones.

Where the paper is weaker is in terms of the flood implications of these long-period waves - the possible role of these waves in enhancing inundation probability is the point on which the paper starts. However, it is not very clear what is their magnitude relative to other water level terms such as surges and waves and how much might they increase total water levels in relative terms? More analysis of when these observed waves might have contributed to floods is needed. There are published papers which do this for similar situations such as Ozsoy et al (2016)

To be published as currently structured the paper needs additional material - maybe a table that sets the magnitude and phase of these meteotsunamis in context with the total water levels that are observed - when might they produce floods and how big is the relative effect?

References

Ozsoy et al (2016). High-frequency sea level variations and implications for coastal flooding: A case study of the Solent, UK. *Continental Shelf Research*, DOI: 10.1016/j.csr.2016.03.021

RESPONSE TO REVIEWERS: (The comments of the reviewers are indicated in black, our response is written in *italic blue* and the changes included in the manuscript in *italic black*)

Reviewer #1 (Formal Review for Authors):

The manuscript presents the presence of meteotsunamis triggered by landfalling hurricanes in observations and idealized simulations. This could be a significant piece of work. However, improvements are needed before it considered acceptable in this highly visible journal.

We thank the reviewer for his/her suggestions. We have tried to address all the points suggested by the reviewer.

(1) It seems that it is already generally known that TCs have the potential to generate meteotsunamis. I wonder if the findings of this work are novel or significant. The authors need to make the following points clear.

Although it is known that TCs might trigger meteotsunamis, the frequency of the occurrence of these types of waves and their generation and propagation mechanisms have not been studied. Mercer et al. (2002) analyzed how tropical storms Jose (1999) and Helene (2000) created barotropic waves with periods of ~10 minutes. However, the waves analyzed by Mercer et al. (2002) are waves created by the entire tropical storm structure. In their study, the low-pressure atmospheric systems associated with the storms propagated at a translation speed of ~30 m/s, which created Proudman resonance over the shelf. Moreover, the authors identified that these storms created wave wakes in the rear side of the storm. These waves are completely different than the types of waves we describe in this study, in which the main triggers of meteotsunamis are not associated with the overall structure of the TC but with atmospheric features with much smaller scales: spiral rainbands. Although Olabarrieta et al. (2017) mentioned several events that created wave elevations larger than 0.4 m, their generation mechanisms were not explained nor analyzed.

All the findings in our study are novel. We have modified the introduction section to emphasize what was known about TC-induced meteotsunamis before our study and to more clearly identify why our study is unique. This text now appears in lines 44-46:

Regardless of the observational evidence that TCs and remnants of TCs have the potential to generate meteotsunamis⁵⁻⁷ that can add to water levels, their frequency and triggering and propagation mechanisms have not been analyzed yet.

This study shows that meteotsunamis frequently occur during TCs and that water level variations in the meteotsunami frequency band should be accounted for when estimating total water levels and impacts. Moreover, we have identified 1) the main atmospheric features that create meteotsunamis and 2) two different types of meteotsunami events, which primarily depend on the characteristics of the rainbands that trigger them. With the idealized numerical simulations, we have demonstrated that an idealized TC propagating over a simplified bathymetry can create the types of meteotsunami events identified in the observations. In our revision of the manuscript, we have included analyses of the relative role of wind stress, atmospheric pressure, and wind wave-induced ocean roughness. Our results indicate that the wind stress divergence is the main driver of TC-triggered meteotsunamis. Moreover, with the idealized simulations, we have analyzed the meteotsunami wave generation and propagation along the wave rays (from 45 m water depth to the coast) in the case of a distant spiral rainband propagating normal to the bathymetry. The analysis shows that these meteotsunami waves are wind driven. The energy transfer from the atmosphere to the ocean is maximized through the Proudman resonance; in our analyzed case, the Proudman resonance occurs at water depths ranging from 20 to 15 m. At shallower water depths, the meteotsunami starts to behave as a free wave and the energy transfer from the atmosphere decreases. Our study also shows how the angle of incidence and the rainband translation speed affects the maximum meteotsunami elevation.

a) Is it known previously that TCs can generate meteotsunamis larger than 0.4 m? *Olabarrieta et al. (2017) mentioned several events that created elevations larger than 0.4 m, but the generation mechanisms were not explained nor analyzed. The reference to the work by Olabarrieta et al. (2017) is included in lines 44-46 as described above.*

b) Is it reported previously that two types of meteotsunamis exist under TCs? *No, the types of generation mechanisms described in this study are novel. We have included language to emphasize the novelty of our discovery in the Discussion section.*

c) Is the magnitude of meteotsunamis reported in this study significant compared to those observed under other conditions? *Yes. In our manuscript revision, we have included the statistical analysis of the maximum elevation of meteotsunamis at the stations that showed the highest meteotsunamis during TCs (Freshwater Canal, Clearwater Beach, Trident Pier, and Naples). We have computed the cumulative distribution of the maximum meteotsunami elevation and have observed that meteotsunamis triggered by TCs with maximum elevations were higher than the 80 % quantile in the eight most extreme cases and were higher than 98.5% in six of those eight (6/8) cases.*

d) Only 4 out 16 analyzed cases have meteotsunamis larger than 0.4 m. Please comment on if you think this is frequent. *We have extended the analysis to include the 1998-2018 Atlantic hurricane seasons and modified the text as follows, lines 98-102:*

In 49 of the 93 analyzed TCs (52 % of the cases), maximum meteotsunami elevations greater than 0.2 m were measured by at least one tide gauge. 19 events (20% of the cases) triggered maximum elevations higher than 0.3 m, and eight of them (8%) produced meteotsunami elevations above 0.4 m (Fig. 1a). The 7 out of 8 of these most extreme meteotsunami events occurred in the Gulf of Mexico (Fig. 1b) and were measured in the right-front and right-rear of the TCs.

19 events (20 % of the cases) triggered maximum elevations higher than 0.3 m, and eight of them (8 % of the cases) produced meteotsunami elevations above 0.4 m. In our opinion, we should describe the results with these percentages.

(2) Lines 72-74 It seems this work did not make any attempt to answer these questions either. *Our results show that TCRs with squall-line-like structure are the main trigger of the highest meteotsunamis, which is one of the main conclusions of our manuscript. In our revised submission, we have reworded and edited our text to clarify and further emphasize this point.*

(3) Lines 212-248 There are a lot of discussions based on the figures not shown in the main content. I suggest either greatly shortening this section or incorporating the related figures to the main content. Particularly, I suggest moving Table 2 in supplementary materials to the main content.

We have edited and reorganized this section to streamline it for our readers. We included a modified version of Supplementary Table 2 to the main text (now Table 1). We have shortened the description of each event and included the probabilistic analysis of the maximum elevation of meteotsunamis (climatology) at the stations that showed the highest meteotsunamis during TCs (Freshwater Canal, Clearwater Beach, Trident Pier, and Naples). We have included a new subsection focused on TC-triggered meteotsunami effects on total water levels.

(4) The analysis of the idealized simulation is a bit cursory. The analysis should be expanded to better reveal the initiation, evolution and characteristics of the modeled meteotsunamis. *The main goal of the idealized case was to show that the capacity of generating meteotsunamis is general to TCs—that even with an idealized case, we are able to see the generation of the two types of meteotsunamis described in the manuscript. We have expanded the analysis by including the analysis of the relative effect of atmospheric pressure vs wind stress and the effect of the ocean wave roughness induced by wind waves. We have analyzed the generation and propagation processes along wave rays for the case of an outer spiral rainband propagating perpendicular to the coast. Our results reveal that wind stress is the main trigger of these types of waves and that the energy transfer from the atmosphere to the ocean increases due to the Proudman resonance process before the meteotsunami waves get unbounded from the atmospheric disturbance and become free waves.*

a) Is wave-coupling necessary in this application? *No, wave-coupling is not necessary. Even without including the increase of the wave roughness due to waves, the model generates meteotsunamis. Nonetheless, the increase of the ocean roughness due to waves increases the elevation of meteotsunamis in ~20% of cases, as is shown in the idealized simulations. This analysis has been included in the new version of the manuscript.*

b) What are the spatial scale and horizontal structure of these waves? *The spatial structures of the waves have been described in the new version of the manuscript (line 359-360).*

Reviewer #2 (Formal Review for Authors):

I found the manuscript very interesting and the topic deserves to be published in a high-rank journal. With an increased availability of high-resolution sea level data, at a minute sampling rate, the research

on extreme sea levels found that meteotsunamis and non-seismic sea level oscillations at the tsunami timescale are contributing substantially to the extremes.

Thank you for the thorough and careful review of our revised manuscript.

Yet, the manuscript lacks a number of important items and analyses, which should be part of it in the case of its publishing in Nature Communications. These are:

1. What is an overall intensity of TC-induced meteotsunamis compared to other types of meteotsunamis? Are TCs minor or major process driving meteotsunamis in the investigated areas? I want to see the analysis of these extreme TC-driven meteotsunami waves versus the climatology of meteotsunami waves at stations where the largest waves are recorded, of course for the period when the high-frequency sea level data is available (and that is at least 10-15 years, I think). By doing that, the overall intensity of TC-induced meteotsunami waves vs. meteotsunami waves induced by other atmospheric processes may be assessed and should be part of the article.

Following the reviewer's suggestion, we have included this analysis in the new version of the manuscript. First, we have analyzed all the meteotsunamis associated with TCs during the 1998-2018 Atlantic Hurricane seasons. Second, we have analyzed all the meteotsunamis that occurred between 1998 to 2018 at those tide gauges that showed the highest meteotsunami elevations. To detect the meteotsunami events, we followed the methodology proposed by Olabarrieta et al. (2017). In the new version of the manuscript, we have included the statistical analysis of the maximum elevation of meteotsunamis at the stations that showed the highest meteotsunamis during TCs (Freshwater Canal, Clearwater Beach, Trident Pier, and Naples). We computed the cumulative distribution of the maximum meteotsunami elevation and observed that in the six most extreme TC-induced meteotsunamis, maximum elevations were higher than 98.5%.

2. I don't see any discussion regarding bathymetry that is in front of tide gauges. Are tide gauges at the open sea or inside harbors?

Most of the gauges are located inside harbors, estuaries, or the intra-coastal region. These details are included in the methods section. The specific characteristics of the stations at which the maximum meteotsunamis were measured are shown in the results section. Naples and Clearwater Beach are located in open waters, while Trident Pier and Fresh Water Canal

Locks are located in channels connecting the open ocean with the inner coastal lagoons. This description has been included in the new version of the manuscript. Lines 136-141:

Most of the NOAA tide gauges along the Gulf of Mexico and Eastern United States are located inside estuaries or in the intra-coastal waterway, sheltered from the direct impact of open ocean waves. Only a few tide gauges are located in open waters, such as at Naples and Clearwater Beach. These are shallow tide gauges, with mean water depths of ~1 m. Freshwater Canal Locks and Trident Pier are deeper gauges, with mean water depths of ~6 m, placed in channels connected to the open ocean.

What is the depth in front of gauges? Is it favorable to generation of meteotsunamis (disturbance speed vs. long ocean waves speed  Proudman resonance?)? Also, I do not see the proof that meteotsunami wave was a solitary wave - this should be proven by mathematics, i.e. that nonlinear and dispersive effects are cancelled in such a wave, so that it maintains its shape over a path. I think that the authors used the wrong terminology here. It looks to me that single atmospheric disturbance is generating the meteotsunami wave though Proudman resonance, and the wind is the dominant mechanism over air pressure. All of these should be analyzed and discussed in the text.

Yes, the reviewer is right. The single -peak waves identified in this study are forced waves, not necessarily solitary waves. We have replaced the term “solitary wave” with “single-peak wave”. We have also extended the idealized numerical modeling section to determine the relative contributions of wind divergence and atmospheric pressure gradient and analyzed the effects of the wave-induced ocean roughness. All this additional analysis is described in the new version of the manuscript. We have also used the idealized model to analyze meteotsunami generation and propagation along specific wave rays for the case of a normal incident outer spiral rainband.

3. Above all of four quoted TC-generated meteotsunamis (Harvey, Irma, Hermine, Michael), the only TS moving parallel with the coast is Irma, during which the series of meteotsunami waves are generated only. Is there any connection between the pathway of a TC (parallel vs. perpendicular to the coast) and meteotsunami waves? I think yes, but the authors should investigate this. Hint: look the study by Sheremet et al. (2016, Natural Hazards, doi: 10.1007/s11069-016-2446-2). The authors may use numerical simulations for doing that.

It is the pathway of the TC rainband (TCR) that matters here, which depends on the pathway of the TC. The TCRs do not necessarily propagate with the same direction as the TC, especially in the case of distant or outer rainbands. We have expanded the idealized TC case and have included the description of the relevance of the direction of propagation of the TCR. In the new Fig 7, we have added a panel which shows how the angle of incidence of a distant TCR affects the magnitude of the meteotsunami. Due to space limitations, we did not include a detailed analysis of the effects of the angle of incidence, although we do mention it. Future studies should include more detailed analysis the effect of the angle of incidence and different bathymetries in relation to TC-induced meteotsunamis.

4. The modelling exercise does not bring anything new to the research, as basically trying to reproduce air pressure and wind effects to the barotropic ocean. For that reason, I would omit it as aggravates the findings, or to use it to prove the underlying physics. As it stands now, it does not provide any new findings regarding physics and processes related to meteotsunami waves. Alternatively, the authors may dig deeply into the physics and go beyond is presented (e.g. following previous comments).

The main aim of the modeling exercise was not to show that pressure and wind perturbations are able to create meteotsunamis; we agree that this is well known as indicated by the reviewer. The main goal of this exercise was to show that even with an idealized TC and over a simplified bathymetry we are able to observe the generation of the types of meteotsunamis identified in our observations. Not only that, but through the numerical simulations, we were able to analyze, in more detail, the generation and propagation processes of the TC-induced meteotsunamis. In addition, this exercise also suggests that the generation of these types of waves might be more common than what was previously thought and are common during TCs. In our opinion, all these points are novel and valuable contributions.

Following the reviewer's suggestion, we have extended the numerical modeling section and included additional analysis about the relative effect of wind stress and atmospheric pressure, the effect of the wind wave-induced ocean roughness, and the generation and propagation mechanisms for the case in which an outer spiral rainband propagates normal to the coast.

The detailed sections on specific hurricanes should be condensed in the manuscript, while details should be moved to supplementary material. Particularly if including new analyses as suggested above.

We have shortened the sections corresponding to each hurricane and included the additional analysis as suggested by the reviewer. We thank the reviewer for his/her suggestions.

Reviewer #3 (Formal Review for Authors):

This is an interesting paper that considers water level variations during tropical cyclones and argues that meteotsunamis are frequently developed. It provides good evidence that this is the case and relatively long-period waves of a few minutes to a few hours are observed. It is also shown that these waves are consistent with generation by pressure and wind fluctuations during tropical cyclones.

Where the paper is weaker is in terms of the flood implications of these long-period waves - the possible role of these waves in enhancing inundation probability is the point on which the paper starts. However, it is not very clear what is their magnitude relative to other water level terms such as surges and waves and how much might they increase total water levels in relative terms? More analysis of when these observed waves might have contributed to floods is needed. There are published papers which do this for similar situations such as Ozsoy et al (2016).

To be published as currently structured the paper needs additional material - maybe a table that sets the magnitude and phase of these meteotsunamis in context with the total water levels that are observed - when might they produce floods and how big is the relative effect?

References

Ozsoy et al (2016). High-frequency sea level variations and implications for coastal flooding: A case study of the Solent, UK. Continental Shelf Research, DOI: 10.1016/j.csr.2016.03.021

Thank you very much for this suggestion. We agree with the reviewer that we might not have emphasized enough the implications of meteotsunamis on the total water levels. In the new version of the manuscript, in the results section, we have added a subsection “TC-induced meteotsunami effects on total water levels” (lines 287-327) dedicated to the analysis of the relative contribution of the meteotsunamis to the total water levels, the phase with respect to the maximum surge, and the ratio between the maximum meteotsunami elevation and the maximum surge measured (at any other tide gauge) during the specific TCs. This subsection includes a new table (Table 2) with all the information related to the contribution to the water levels.

REVIEWERS' COMMENTS:

Reviewer #1 (Remarks to the Author):

The manuscript is substantially improved after the revision. The authors' responses to my comments are overall satisfactory. I have two minor comments.

1) The authors gave a detailed response to my first comment. These details need to be reflected in the manuscript. Their statement on lines 44-46 is too brief. Please consider add these details in the Discussion section.

2) Lines 469-471 The effect of wave-coupling on surface roughness is not well established in my understanding. The 18-26% increase of meteotsunami magnitude due to wave coupling could only be true in this particular model configuration. I think the authors should make this point clear.

Reviewer #2 (Remarks to the Author):

I really appreciate a substantial effort invested by authors to make the manuscript much better than in its initial version. No further comments to post, the manuscript is of enough quality to be published in Nature Communications.

Ivica Vilibic

Reviewer #3 (Remarks to the Author):

Meteotsunamis triggered by tropical cyclones

I reviewed an earlier version of this paper and I must commend the authors on the positive manner in which they have reacted to all the reviews they received. The paper has been thoroughly revised and all the reviewers have been addressed to my satisfaction. They have convinced me that meteotsunamis under tropical cyclones occur frequently and can be a significant contribution to total water levels under some conditions. Therefore, I am happy to see this paper published in Nature Communications.

My big comment would be to add at the end of the Discussion a few thoughts on the implications of this work for coastal science and coastal flood management practice. They state on line 487/488 "meteotsunami effects must be incorporated into future models for accurate flood-risk forecast and assessment". This is a bit vague and it would be good to have a little more structure and one or two sentences or better an additional paragraph on implications.

Building on this suggestion, on lines 55/56 they say "Due to the coarse spatial (> 3 km) and temporal resolutions (> 1-3 hours) of the atmospheric fields usually available to force storm surge models, current forecasting systems used to simulate water levels during TCs are ill-equipped to capture and model meteotsunamis. ". This raises the question of what atmospheric fields are required to model meteotsunamis?

There are many possible applications of this knowledge – two spring to mind as especially important. One is on return periods of these meteotsunami events and how they might contribute to our understanding of extreme water levels for design purposes. So we define a 100 year water level for flood insurance purposes – does this make it higher? Similarly, what is required for short-term meteotsunami forecasts as part of hurricane warning. And is this feasible with our current knowledge, models and data availability? Please expand on these points briefly.

Two smaller points.

Line 32,33 Sea-level rise is essentially inevitable – changes to tropical cyclones is much more uncertain – different levels of confidence need to be distinguished.

Line 78 onwards – Text is more Methods more than Results – not saying it is not needed but is it titled correctly? I leave this to the editors to consider.

RESPONSE TO REVIEWERS: (The comments of the reviewers are indicated in black, our response is written in *italic blue* and the changes included in the manuscript in *italic black*)

Reviewer #1 (Remarks to the Author):

The manuscript is substantially improved after the revision. The authors' responses to my comments are overall satisfactory. I have two minor comments.

Thank you very much for your comments, suggestions and the time spent reviewing this manuscript.

1) The authors gave a detailed response to my first comment. These details need to be reflected in the manuscript. Their statement on lines 44-46 is too brief. Please consider add these details in the Discussion section.

Following the suggestion of the reviewer, we have added the following paragraph to the beginning of the discussion section:

The potential of TCs to trigger meteotsunamis was demonstrated by Mercer et al.⁵, who analyzed how tropical storms Jose (1999) and Helene (2000) created barotropic waves with periods of ~10 minutes across the Grand Banks of Newfoundland. The low-pressure atmospheric systems associated with these two tropical storms propagated at a translation speed of ~30 m s⁻¹, which created Proudman resonance over the shelf and forced wave wakes in the rear side of the storms. Unlike the barotropic waves described by Mercer et al.⁵, the meteotsunamis identified and analyzed in this study are not associated to the whole structure of the storm but rather with atmospheric features with much smaller scales: the spiral rainbands (TCRs).

2) Lines 469-471: The effect of wave-coupling on surface roughness is not well established in my understanding. The 18-26% increase of meteotsunami magnitude due to wave coupling could only be true in this particular model configuration. I think the authors should make this point clear.

Yes, the reviewer is right. To make clear this point we have rephrased the sentence:

In the idealized meteotsunami event analyzed herein, wind wave induced ocean surface roughness increased the magnitude of wind shear stress and this translated into an increase of the meteotsunami magnitude of about 18-26%.

Reviewer #2 (Remarks to the Author):

I really appreciate a substantial effort invested by authors to make the manuscript much better than in its initial version. No further comments to post, the manuscript is of enough quality to be published in Nature Communications.

Ivica

Thank you very much Ivica for your time and efforts; your comments and suggestions were key to improve our analysis.

Reviewer #3 (Remarks to the Author):

Meteotsunamis triggered by tropical cyclones: I reviewed an earlier version of this paper and I must commend the authors on the positive manner in which they have reacted to all the reviews they received. The paper has been thoroughly revised and all the reviewers have been addressed to my satisfaction.

We appreciate your time, effort and the comments about the total water levels.

They have convinced me that meteotsunamis under tropical cyclones occur frequently and can be a significant contribution to total water levels under some conditions. Therefore, I am happy to see this paper published in Nature Communications.

My big comment would be to add at the end of the Discussion a few thoughts on the implications of this work for coastal science and coastal flood management practice. They state on line 487/488 “meteotsunami effects must be incorporated into future models for accurate flood-risk forecast and assessment”. This is a bit vague and it would be good to have a little more structure and one or two sentences or better an additional paragraph on implications.

Building on this suggestion, on lines 55/56 they say “Due to the coarse spatial (> 3 km) and temporal resolutions (> 1-3 hours) of the atmospheric fields usually available to force storm surge models, current forecasting systems used to simulate water levels during TCs are ill-equipped to capture and model meteotsunamis. “. This raises the question of what atmospheric fields are required to model meteotsunamis?

There are many possible applications of this knowledge – two spring to mind as especially important. One is on return periods of these meteotsunami events and how they might contribute to our understanding of extreme water levels for design purposes. So we define a 100 year water level for flood insurance purposes – does this make it higher? Similarly, what is required for short-term meteotsunami forecasts as part of hurricane warning. And is this feasible with our current knowledge, models and data availability? Please expand on these points briefly.

We have included a new paragraph at the end of the discussion section as follows:

Because of the inherent ability of meteotsunamis to increase coastal damage and erosion, meteotsunami effects must be incorporated into future models for accurate flood-risk forecast and assessment. For example, the design of coastal structures and coastal-flood insurances are based on the total water levels estimated for a given return period. The total water levels for a given return period could be under-estimated if meteotsunamis are not considered. In regards to flood-risk forecasts during hurricane warnings, the consideration of meteotsunamis could improve the accuracy of the forecast. However, this would require the atmospheric models to accurately forecast specific TCRs and the hydrodynamic models to be forced with high temporal (~5 minutes) and spatial (~500 m) resolution wind and sea level atmospheric pressure fields. Despite considerable advancements in knowledge and modeling capabilities of the inner structure of TCs, forecasting specific TCRs is still challenging. Moreover, feeding the hydrodynamic models with higher resolution atmospheric fields could result in a prohibitive increase in computational cost. Further research on meteotsunami forecasting is needed to overcome these limitations.

Two smaller points.

Line 32,33 Sea-level rise is essentially inevitable – changes to tropical cyclones is much more uncertain – different levels of confidence need to be distinguished.

The reviewer is referring to the following sentence:

As human populations along the coast continue to increase and sea levels rise, tropical cyclones will likely result in enhanced coastal impacts in the future²⁻⁴.

With this sentence, we are not saying that we expect the tropical cyclones to change; the vulnerability (and therefore the risk and impacts) of coastal communities will increase because water levels are higher and because there will be more people living along the coast. It is implicit that sea level rise is inevitable; we don't mention anything about changes in TC patterns because this is more uncertain. We decided to leave this sentence as it is.

Line 78 onwards – Text is more Methods more than Results – not saying it is not needed but is it titled correctly? I leave this to the editors to consider.

Following the suggestion of the editor, we haven't moved these two paragraphs.